# Design and Measurement of a 0.67 THz Biased Sub-Harmonic Mixer

**Guangyu Ji [1,2], Dehai Zhang [1,*], Jin Meng [1], Siyu Liu [1,2] and Changfei Yao [3]**

[1] CAS Key Laboratory of Microwave Remote Sensing, National Space Science Center,
Chinese Academy of Sciences, Beijing 100190, China; guangyuji1@163.com (G.J.); mengjin@mirslab.cn (J.M.);
liusiyu16@mails.ucas.edu.cn (S.L.)

[2] University of Chinese Academy of Sciences, Beijing 100049, China

[3] School of Electronic and Information Engineering, Nanjing University of Information
Science and Technology, Nanjing 210044, China; yaocf1982@163.com

* Correspondence: zhangdehai@mirslab.cn; Tel.: +86-010-6258-6483

**Abstract:** To effectively reduce the requirement of Local Oscillator (LO) power, this paper presents the design and measurement of a biased sub-harmonic mixer working at the center frequency of 0.67 THz in hybrid integration. Two discrete Schottky diodes were placed across the LO waveguide in anti-series configuration on a 50 μm thick quartz-glass substrate, and chip capacitors were not required. At the driven of 3 mW@335 GHz and 0.35 V, the mixer had a minimum measured Signal Side-Band (SSB) conversion loss of 15.3 dB at the frequency of 667 GHz. The typical conversion loss is 18.2 dB in the band of 650 GHz to 690 GHz.

**Keywords:** bias; sub-harmonic mixer; anti-series; Schottky diode; conversion loss

## 1. Introduction

Terahertz usually refers to the frequency band between 0.1 THz to 10 THz. In recent years, there is an urgent demand for receivers operating at terahertz frequency in radio astronomy, planetary exploration, and atmospheric remote sensing [1]. As is well known, it is difficult to produce high Local Oscillator (LO) power in sub-millimeter and Terahertz frequency for lacking power amplifiers [2,3]. Harmonic mixers are widely used in terahertz heterodyne receivers due to the advantage of reducing the LO frequency. Sub-harmonic mixing and fourth harmonic mixing are the most-utilized mixing methods, and conversion loss increases with the number of mixing times [4].

At the frequency below 600 GHz, unbiased sub-harmonic mixers can achieve good noise performance. In Reference [5], sub-harmonic mixers operating at 183 GHz and 366 GHz are designed based on 3.7 μm thick GaAs membrane. The Double Side-Band (DSB) conversion loss and noise temperature of the 183 GHz sub-harmonic mixer are 4.9 dB and 608 K respectively, and the 366 GHz mixer is 6.9 dB and 1220 K. In Reference [6], a sub-harmonic mixer is designed working at 190–240 GHz using a discrete Schottky diode. The DSB noise temperature is lower than 1500 K and the DSB conversion loss is less than 10 dB at the frequency band. In addition, some similar unbiased sub-harmonic mixer designs can be found in Reference [7–10].

The output power of the LO sources decreases with increasing frequency, especially when the operating frequency reaches 0.6 THz and above [11,12]. So, it is urgent to design biased sub-harmonic mixers to reduce the requirement for LO power. At present, biased sub-harmonic mixers working at 585 GHz [13], 874 GHz [14,15], 1.2 THz [16,17], and 1.2 THz [18] are designed and reported based on advanced GaAs membrane film process or frameless architecture. The biased mixers mentioned above are all based on monolithic integration technology, where the on-chip capacitor is required. But, the thickness of the GaAs substrate is less than 5 μm, which is easy to bend and expensive.

It is valuable to research the biased sub-harmonic mixer in hybrid integration to solve practical engineering problems of low LO power. Because of the big size and poor performance of discrete chip capacitors in terahertz frequency, it is unable to achieve good performance to mixers in hybrid integration of anti-parallel configuration. In this paper, the biased hybrid integration scheme is adopted. The scheme is theoretically derived for the first time and compared with the anti-parallel structure to analyze the advantages and disadvantages. Two discrete Schottky diodes are biased by DC voltage and placed across the LO waveguide in anti-series configuration without chip capacitors. The mode of the LO and RF signals are orthogonal in mixing, so the LO and RF ports are highly isolated. The bias voltage is feed from the IF port and separated by a bias-T.

Section 2 illustrates the theoretical analysis and comparison of different mixing topologies. Section 3 presents the detailed architecture, simulation process, and simulation results of the 0.67 THz biased sub-harmonic mixer. The measurement platform and results of the mixer are depicted in Section 3 at the same time. Section 4 gives the discussion and comparison in simulation, measurement results. Finally, the conclusion is presented in Section 5.

## 2. Comparison and Analysis of Different Mixing Topologies

Three different mixing topologies are shown in Figure 1. Figure 1a,b presents two anti-parallel circuit topologies that are commonly used in terahertz sub-harmonic mixers, including biased and unbiased mixing. The currents direction of the primary, secondary and tertiary harmonics of the LO and RF signals of the two diodes are illustrated in Figure 1. The total mixing current contains frequency terms $f = m f_{RF} \pm n f_{LO}$ listed in Table 1, where m and n is integer. To sub-harmonic mixers, the IF signal is the only concerned frequency which can be expressed as $f_{IF} = |f_{RF} - 2 f_{LO}|$.

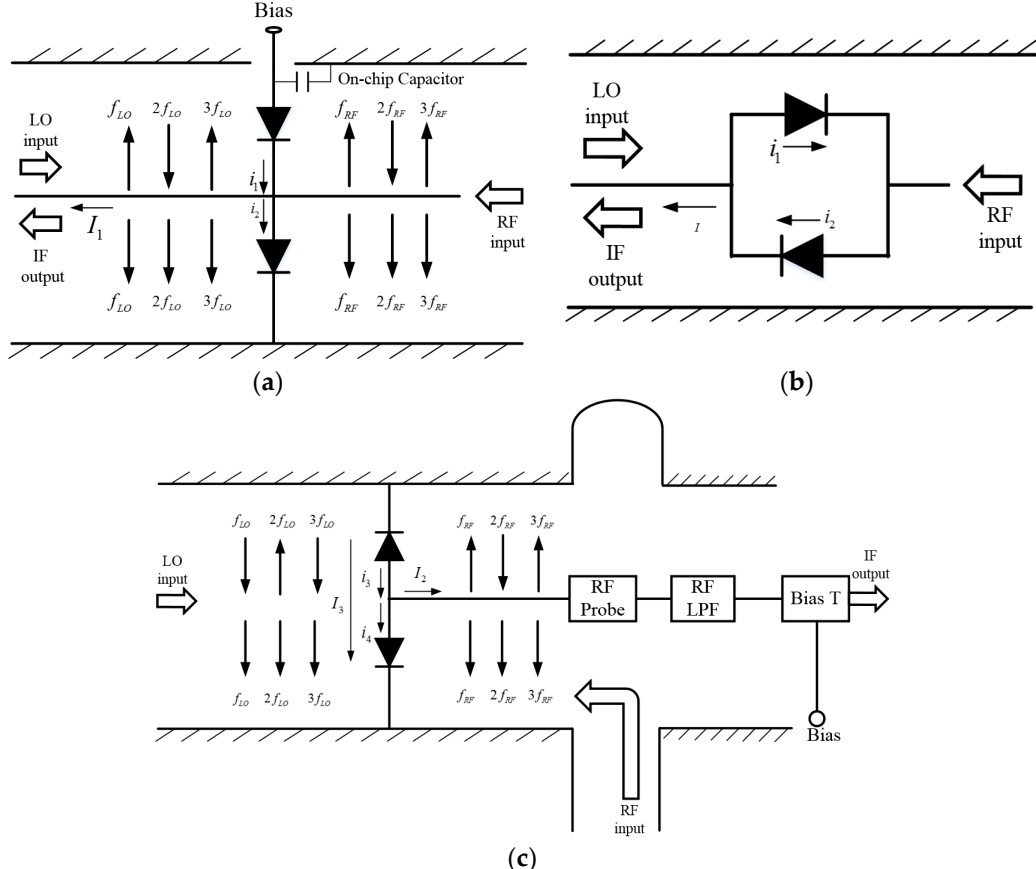

**Figure 1.** (**a**) Topology of biased anti-parallel; (**b**) topology of unbiased anti-parallel; (**c**) the topology of biased anti-series.

**Table 1.** Mixing products of anti-parallel and anti-series diodes configuration.

| | Output Signal ($f=mf_{RF}\pm nf_{LO}$) | | | |
|---|---|---|---|---|
| | m:odd n:odd | m:even n:even | m:odd n:even | m:even n:odd |
| Anti-Parallel $I_1$ | × | × | √ | √ |
| Anti-Series $I_2(I_3)$ | × (√) | √ (×) | √ (×) | × (√) |

Sub-harmonic mixers using the topology shown in Figure 1a typically are based on monolithic integration due to the one on-chip capacitor required. It has the same equivalent circuit with the topology in Figure 1b, which is the anti-parallel mixing configuration. The RF and LO signals are feed to diodes in quasi-Transverse Electromagnetic (TEM) mode. In Figure 1a,b, $i_1$ and $i_2$ have the same frequency components. The only output current I can be expressed as $I_1 = i_2 - i_1$. It means that there is output only when $i_1$ and $i_2$ are in the opposite phase. The currents through the diode junctions can be written as

$$\begin{cases} i_1 = I_s(e^{-\alpha V} - 1) \\ i_2 = I_s(e^{\alpha V} - 1) \end{cases} \tag{1}$$

where $I_s$ is the reversed saturation current; $\alpha$ is the slope parameter ($\alpha = \frac{q}{nkT}$), where $k$ is the Boltzmann constant, n is the ideality factor, and T is the operating temperature of the diode.

The conductance of each diode junction is

$$\begin{cases} g_1 = \frac{di_1}{dV} = -\alpha I_s e^{-\alpha V} \\ g_2 = \frac{di_2}{dV} = \alpha I_s e^{\alpha V} \end{cases} \tag{2}$$

The mixing current of the two diode junctions is written as

$$\begin{cases} i_1 = g_1(v_{RF}cos\omega_{RF}t + v_{LO}cos\omega_{LO}t) \\ i_2 = g_2(v_{RF}cos\omega_{RF}t + v_{LO}cos\omega_{LO}t) \end{cases} \tag{3}$$

The total mixing current of the anti-parallel diode pair is

$$\begin{aligned} I_1 = i_1 - i_2 &= 2\alpha I_s \sinh(v_{LO}cos\omega_{LO}t) * v_{RF}cos\omega_{RF}t + 2\alpha I_s \cosh(v_{LO}cos\omega_{LO}t) * v_{LO}cos\omega_{LO}t \\ &= A\cos\omega_{RF}t + B\cos\omega_{LO}t + C\cos 3\omega_{LO}t + D\cos 5\omega_{LO}t \\ &\quad + E\cos(2\omega_{LO}t + \omega_{RF}t) + F\cos(2\omega_{LO}t - \omega_{RF}t) \\ &\quad + G\cos(4\omega_{LO}t + \omega_{RF}t) + H\cos(4\omega_{LO}t - \omega_{RF}t) \\ &\quad + \cdots + X\cos(m\omega_{LO}t + n\omega_{RF}t) + \cdots \end{aligned} \tag{4}$$

As listed in Table 1, the total mixing current only contains frequency terms $f = mf_{RF} \pm nf_{LO}$, where $m + n$ is odd [19]. The anti-parallel configuration can suppress half of the mixed signal called balanced structure.

The topology of the mixer designed in this paper is presented in Figure 1c. Two Schottky diodes are in anti-series across the LO waveguide [20]. Thus, the two diodes are turned on and off alternately along with the LO signal in TE10 mode. The RF signal is applied to diodes in quasi-TEM mode, where the RF probe is used to transfer TE10 mode to quasi-TEM mode. The mixing current $I_2$ along the microstrip line is $I_2 = i_4 - i_3$ and outputs only when the phase difference of $i_3$ and $i_4$ is π. The output current $I_2$ is expressed as $I_2 = i_4 + i_3$, and output when $i_3$ and $i_4$ are in the same phase.

The currents of junctions can be written as

$$\begin{cases} i_3 = -I_s(e^{-\alpha V} - 1) \\ i_4 = I_s(e^{\alpha V} - 1) \end{cases} \tag{5}$$

The time-varying conduction is

$$\begin{cases} g_3 = \frac{di_3}{dV} = \alpha I_s e^{-\alpha V} \\ g_4 = \frac{di_4}{dV} = \alpha I_s e^{\alpha V} \end{cases} \tag{6}$$

The mixing current of diode junctions is

$$\begin{cases} i_3 = g_3(v_{LO}cos\omega_{LO}t - v_{RF}cos\omega_{RF}t) \\ i_4 = g_4(v_{LO}cos\omega_{LO}t + v_{RF}cos\omega_{RF}t) \end{cases} \tag{7}$$

The mixing current along the microstrip line is

$$\begin{aligned} I_2 = i_4 - i_3 &= 2\alpha I_s \cosh(v_{LO}cos\omega_{LO}t) * v_{RF}cos\omega_{RF}t \\ &+ 2\alpha I_s \sinh(v_{LO}cos\omega_{LO}t) * v_{LO}cos\omega_{LO}t \end{aligned} \tag{8}$$

The mixing current of along the LO waveguide is

$$\begin{aligned} I_4 = i_4 + i_3 &= 2\alpha I_s \sinh(v_{LO}cos\omega_{LO}t) * v_{RF}cos\omega_{RF}t \\ &+ 2\alpha I_s \cosh(v_{LO}cos\omega_{LO}t) * v_{LO}cos\omega_{LO}t \end{aligned} \tag{9}$$

As listed in Table 1, half of the frequency components can be prevented from output to the microstrip line. However, part of the mixing signal leaks from the LO waveguide and cannot be reused. The leakage results in about 3 dB increment of the conversion loss compared with the anti-parallel mixers in principle. So, the topology is not a balanced structure in Figure 1c.

The advantage of the topology in Figure 1c is that biased mixing can be achieved without using on-chip capacitors, which is a big advantage to sub-harmonic mixers in hybrid integrated in terahertz.

## 3. Mixer Design

### 3.1. Mixer Architecture

Figure 2 shows the overall passive circuit structure built in a high frequency structure simulator (HFSS) of the 0.67 THz mixer designed. The LO signal is fed by the WR2.8 rectangular waveguide (711 μm × 356 μm) and reduced the height to 150 μm, while the RF is WR1.5 (381 μm × 191 μm) and reduce the height to 120 μm. Two planar channel Schottky diodes are placed in anti-series across the LO waveguide.

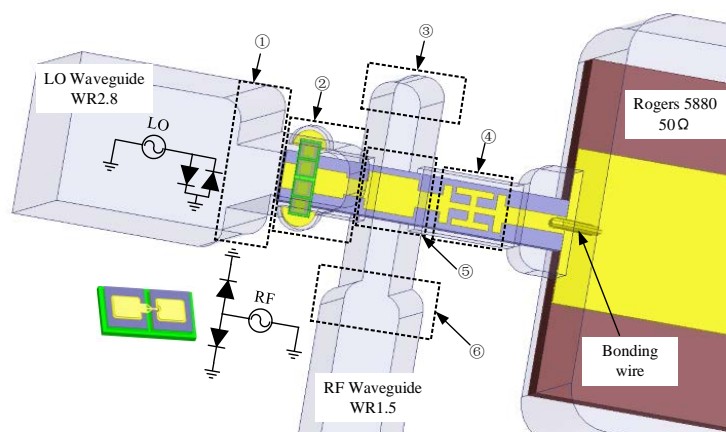

**Figure 2.** The overall passive circuit modeled in the high frequency structure simulator (HFSS). ① Local Oscillator (LO) reduction waveguide. ② Schottky diode pair with grounded ears. ③ RF waveguide backs short. ④ IF LPF. ⑤ RF probe. ⑥ RF reduction waveguide.

The RF probe is used to coupling the RF signal to the microstrip planar circuit which transfers the TE10 mode to quasi-TEM mode. Due to the orthogonality of TE10 mode and TEM mode, the LO port and RF port are highly isolated. Only one IF LPF (low pass filter) is needed to extract the IF signal and reflect additional harmonics to the diode pair. Due to the difference of mode, the diode pair presents parallel to the LO signal and anti-series to the RF signal. The Rogers 5880 substrate is used as the transition between SMA and quartz substrate.

The mixer has been carefully considered in the following aspects:

- To minimize the size of the mixer, an external Bias-T is used instead of designing inside.
- The cathode pads of the two diodes are directly placed on the pre-designed grounded ears as shown in Figure 3 which can short the LO and RF signal. At the same time, the loop is provided for the IF signal and bias voltage.
- To improve the stability of fabrication and assembly, the Rogers 5880 substrate is used as the transition board to reduce the length of the quartz-glass.
- To enhance the stability of the diode assembly, the shield microstrip line is chosen instead of the suspended microstrip line. The size of the circuit channel is 150 μm × 100 μm which can guarantee the single-mode transmission of the RF signal.

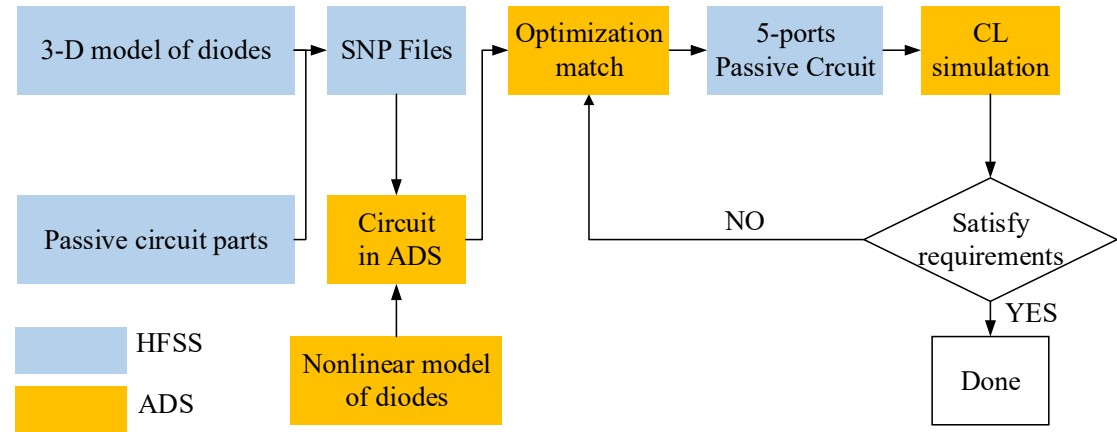

**Figure 3.** Simulation flow of the 0.67 THz biased mixer.

### 3.2. Mixer Simulation

Two single-anode Schottky diodes are utilized which are the SD1G2 series produced by Teratech Components Ltd. The diode is based on planar channel structure and the cut-off frequency is about 14 THz which can fully meet the requirement of the 0.67 THz mixer design. Table 2 lists the relative parameters of the Schottky diode.

**Table 2.** Parameters of the Schottky diode.

| Cj0 | Rs | Cut-Off Frequency | Overall Size (L × W × H) |
| --- | --- | --- | --- |
| 0.95 fF | 11.5 Ω | 14 THz | 90 μm × 50 μm × 12 μm |

The simulation flow of the 0.67 THz sub-harmonic mixer is presented in Figure 3, which combines the HFSS and Advanced Design Software (ADS). Firstly, in the structure shown in Figure 2, the passive circuit of the mixer is divided into 6 parts. The corresponding scatter-parameters are calculated and exported to SNP files. Second, the complete mixer circuit is built in the ADS, and the harmonic balance algorithm is used to calculate the conversion loss and optimize the matching circuit. Third, the 5-ports overall passive circuit is modeled and simulated in the HFSS and combines the nonlinear diode model in the ADS to verify the final performance. The design is an iterative process.

The circuit in ADS is presented in Figure 4, which is based on the overall optimization method. The SNP 1 to SNP 6 is the corresponding S-parameters of parts in Figure 2. The biased T junction is used to separate the DC voltage and the IF signal and has no insertion loss, which is formed by one ideal capacitor and inductor. The mixing principle is based on the nonlinearity of the Schottky junction which is controlled by the LO signal. According to [21], the LO power is about 6 mw to unbiased sub-harmonic mixers at 670 GHz.

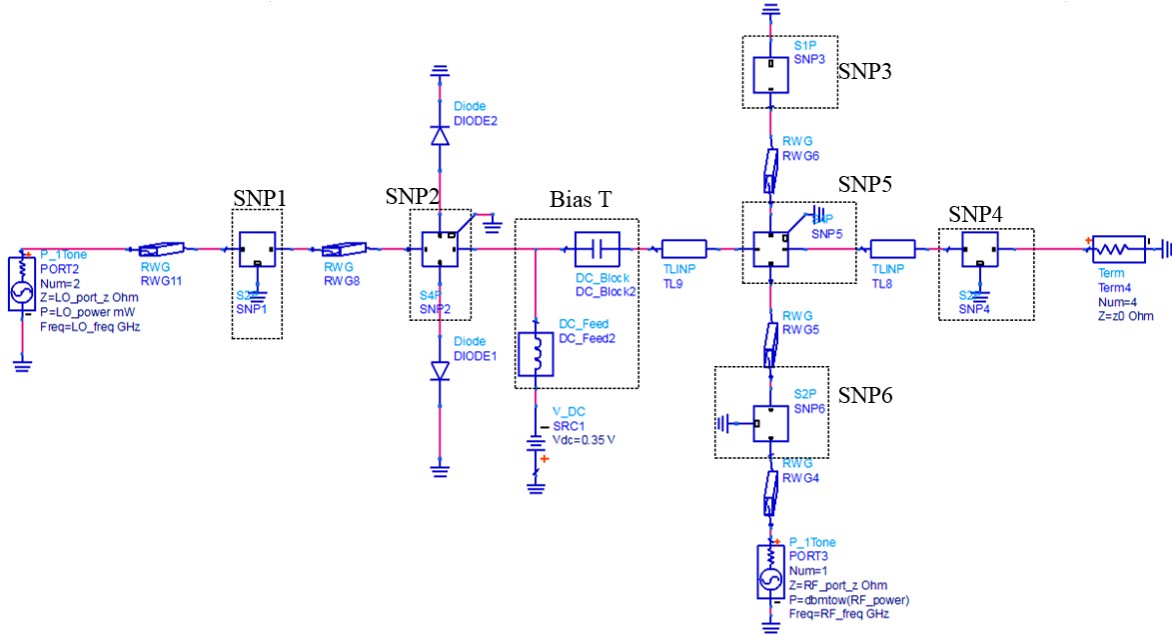

**Figure 4.** Overall circuit in Advanced Design Software (ADS).

Figure 5 shows the curve of conversion loss under different LO power and DC voltage when the LO and RF frequencies are set to 335 GHz and 671 GHz, respectively. When the LO power is fixed at 2 mw and bias voltage ranges from 0.1 V to 0.9 V, the minimum conversion loss is achieved around the voltage of 0.35 V. Since the driving power is insufficient in the range of 0 to 0.35 V, the conversion loss decreases with the bias voltage increases. However, conversion loss increases in the interval of 0.35 V to 1 V because of the nonlinearity reduction of diodes with the increasing of the bias voltage. When the LO power changes, the phenomenon is similar to the above. If a combination of lower LO power (<2 mw) and higher bias voltage (>0.35 V) is used, the optimal conversion loss cannot achieve because of the dynamic range of the diode junction caused by the LO signal is low.

Figure 6 shows the simulation results when the LO power is 2 mw and the bias voltage is 0.35 mV. The conversion loss is from 10 dB to 12 dB when the RF frequency is range from 653 GHz to 710 GHz. When the LO frequency is varied from 330 GHz to 340 GHz, the change in conversion loss is less than 1 dB. This indicates that the 0.67 THz biased mixer has good RF and LO bandwidth characteristics.

Figure 7 shows the simulated isolation between the RF, LO, and IF ports. Due to the orthogonality of TE10 mode and TEM mode, the isolation of LO port to the RF port is above −50 dB between 300 GHz to 400 GHz, and the isolation of RF to LO is above −29 dB between 650 GHz to 720 GHz. Due to the IF filter, the RF power cannot leak to the IF port, and its isolation is above −18 dB from 650 GHz to 710 GHz.

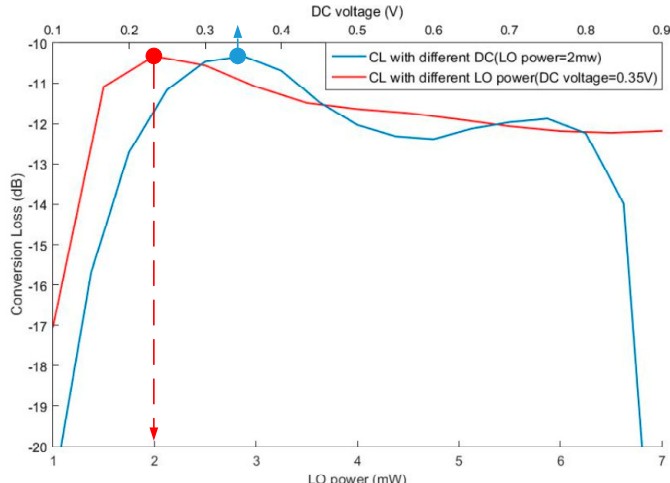

**Figure 5.** Conversion loss with different LO power and DC voltage.

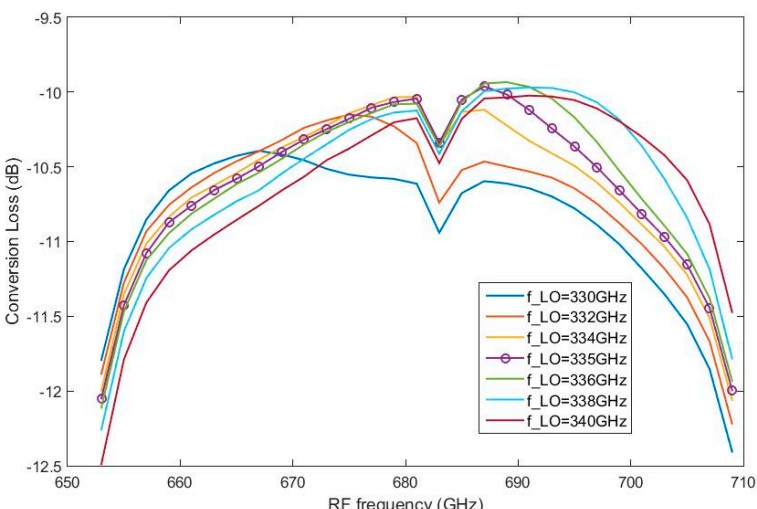

**Figure 6.** Conversion loss with different LO frequencies.

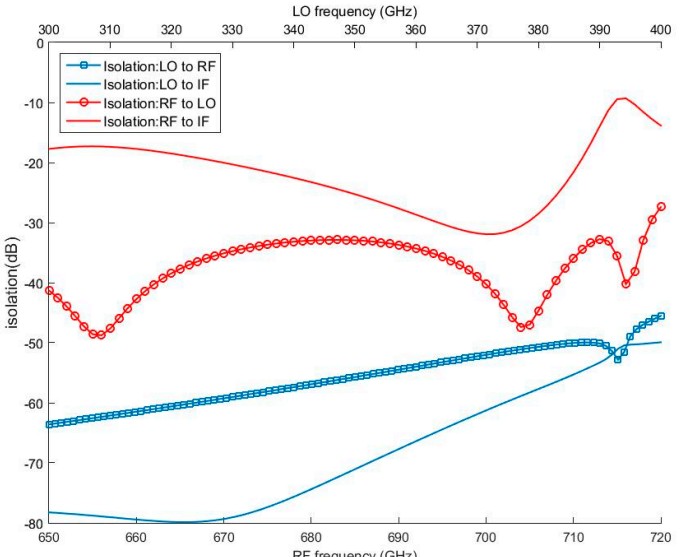

**Figure 7.** Simulated isolation of RF, LO, and IF ports.

### 3.3. Mixer Fabrication and Measurement

The cavity of the designed 0.67 GHz biased mixer is made of brass material, and the entire surface is gold plated. The mixer integrates two UG387 flanges for LO and RF waveguides and a female SMA connector for the IF signal. To facilitate assembling, the cavity is cut from the center of the E plane of the waveguide into two parts and locked by screws. As shown in Figure 8, the quartz-glass substrate and diodes are fixed by conductive adhesive, and the Rogers 5880 substrate is glued by tin solder.

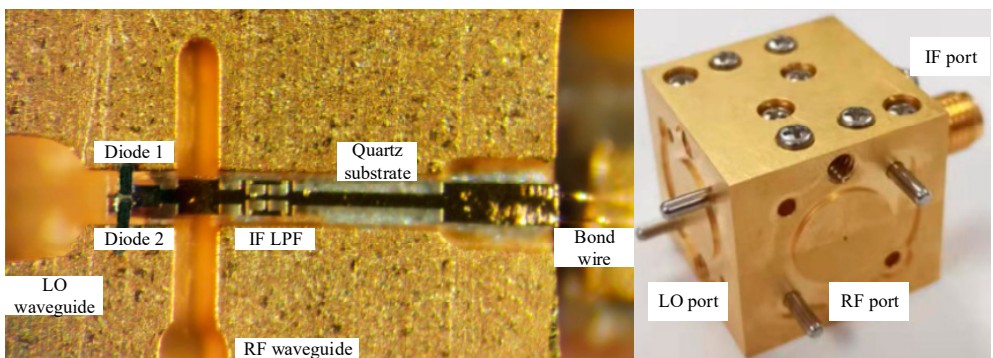

**Figure 8.** Circuit in the lower block of the mixer; assembled mixer.

Figure 9 shows the test diagram of the mixer. The RF and LO signals are generated by two different links, and the power and spectrum of the IF signal are measured by a spectrum analyzer. An external Bias T is used to separate the DC and IF signals. The LO signal is generated by a multiplier chain which is composed of a signal generator, W band multiplier, W band PA, and 330 GHz doubler. The RF signal is and generated by a signal generator and * 54 multiplier module produced by VDI.

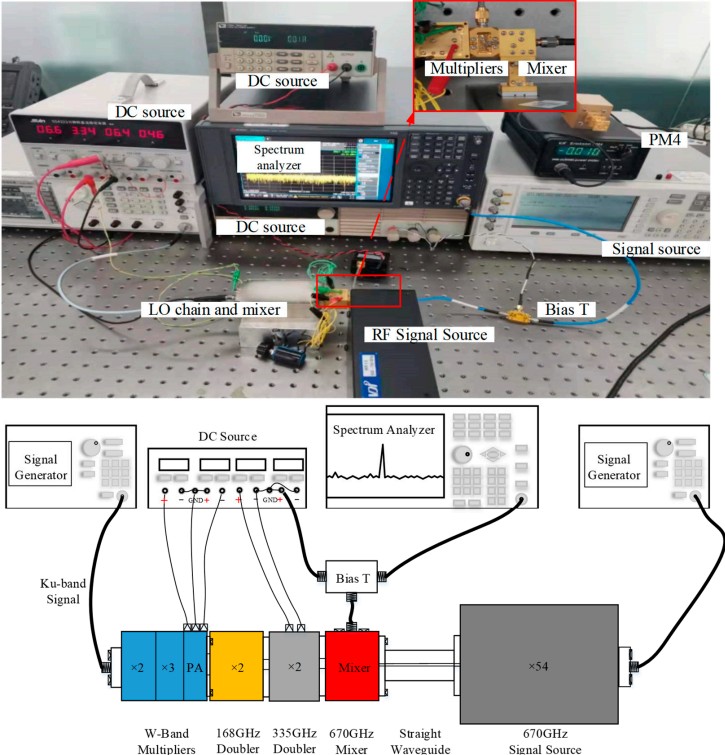

**Figure 9.** Test bench and measurement diagram of the mixer.

Before measuring the conversion loss, the respective output power of the LO and RF chains needs to be measured and calibrated by a PM4 power meter. The maximum LO power generated by the LO chain is about 13 mW between 330 GHz and 340 GHz. The RF signal power is between −18 dBm to −15 dBm from 600 GHz to 700 GHz.

Figure 10 presents the Signal Side-Band (SSB) conversion loss versus RF frequency when the LO is fixed at 2 mw@335 GHz. The conversion loss shown has been corrected for attenuation of cables and the insertion loss of the Biased T. The best measured SSB conversion loss is 15.3 dB@667 GHz. In the RF band of 650 GHz–690 GHz, the conversion loss is below 20 dB, besides the frequency of 679 GHz and 685 GHz. The typical SSB conversion loss is 18.2 dB.

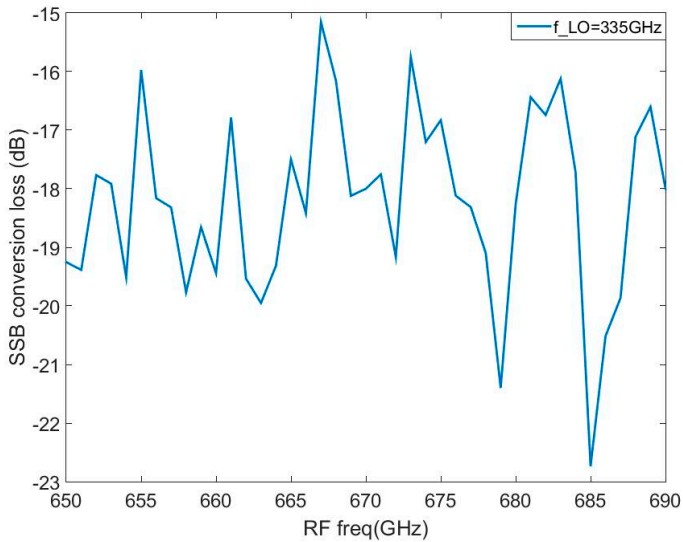

**Figure 10.** Measured Signal Side-Band (SSB) conversion loss.

## 4. Discussion

As shown in Figure 10, the curve of the SSB conversion loss has a large jitter in the band of 670 GHz to 690 GHz than in 650 GHz to 670 GHz. The phenomenon can be explained as follows. First, the RF power is measured by the PM4 power meter which is a thermal power meter that calculates the total power of input signals. While the output RF power ranges from −20 dBm (10 mW) to −18 dBm (15.8 mW) in 670 GHz to 690 GHz, which cannot be measured accurately. The 3 mW to 5 mW measurement error of the PM4 power meter is normal which may lead conversion loss error of 2 dB. Second, the RF source utilized is a frequency multiplier module (* 54) which has many harmonic components that affect the RF power.

As can be seen from Figures 6 and 10, there is a conversion loss gap between simulation and measurement. The gap is caused by the following reasons. First, the nonlinear model of the Schottky diode junction utilized in simulation is provided by the ADS software, which is a P-N junction model but not Schottky diode junction (metal-semiconductor junction). When the working frequency reaches 0.6 THz and above, the current saturation in the diode junction and planar structure can increase the conversion loss [21]. It is urgent to modify the diode junction model in the simulation. Second, the assembling error of Schottky diodes and the substrate deteriorates the performance, which is caused by the thickness of conductive adhesive and the alignment of the diodes. In the assembling, the alignment error between the two diodes is about 20 μm, which leads to unbalanced mixing and deteriorate the conversion loss. Besides, the conductivity of conductive adhesive is not ideal and cannot be accurately characterized in simulation, as the conductive adhesive is the self-made mixture of epoxy and silver.

Table 3 shows several reported sub-harmonic mixers and receivers working around 600 GHz. The mixer reported in Reference [10,22] are all based on anti-parallel Schottky diode pair and hybrid integration. Thus, the LO power is higher than the mixer designed in this paper. The 0.67 THz

biased sub-harmonic mixer has better conversion loss than the mixer in [21]. The mixer reported in Reference [13] is based on advanced membrane monolithic integration technology and anti-parallel architecture. Thus, it has better conversion loss than the mixer design.

**Table 3.** Summary of published terahertz mixer working around 600 GHz.

| Frequency (GHz) | Biased | LO Power (mw) | Conversion Loss (dB) | Structure | Reference |
|---|---|---|---|---|---|
| 638–715 | No | 2–8 | 8.2–12 (DSB) | Hybrid | [10] |
| 520–590 | Yes | | 10.6–11.7 (DSB) | Monolithic | [13] |
| 660–710 | No | 6 | 13–20 (DSB) | Hybrid | [22] |
| 650–690 | Yes | 3 | Optimum: 15.3 (SSB) Typical: 18.2 dB | Hybrid | This work |

Compared with those mixers, the 0.67 THz biased sub-harmonic mixer has disadvantages and advantages as follows. In terms of conversion loss, the anti-series topology utilized has intrinsically defective in suppressing harmonics compared with anti-parallel topology, as analyzed in Section 2. But the anti-series topology can be used to biased mixers in the hybrid integration that effectively reduce the LO power. The performance can be improved in two ways in simulation. First, the Voltage Standing Wave Ratio (VSWR) of the LO port needs to be further improved to reduce the LO power. Second, the anti-series Schottky diode pair can be used to replace the two discrete diodes. The diode pair can eliminate the alignment error of the discrete diodes in assembling.

## 5. Conclusions

A 0.67 THz biased sub-harmonic mixer in hybrid integration has been designed and measured based on an anti-series Schottky diode placed across the LO waveguide. The circuit topology utilized is detailed, analyzed and compared with traditional anti-parallel configurations. The mixer design can realize the biased mixing structure without using discrete chip capacitors, and high isolation between RF, LO, and IF ports. The measured data shows the optimum SSB conversion loss is 15.3 dB at 667 GHz. In the RF frequency band of 650 GHz to 690 GHz, the typical value of conversion loss is 18.2 dB. The mixer design effectively decreased the LO power in the hybrid integrated Schottky diode-based sub-harmonic mixer. At the same time, the 0.67 THz biased sub-harmonic mixer has great prospects in ice cloud detection and planetary exploration.

**Author Contributions:** Conceptualization, methodology, software, and writing, G.J., D.Z., J.M., and C.Y.; investigation, S.L.; visualization, G.J. All authors have read and agreed to the published version of the manuscript.

**Funding:** This research received no external funding.

**Conflicts of Interest:** The authors declare no conflict of interest.

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
