# Peer review of "Design and Measurement of a 0.67 THz Biased Sub-Harmonic Mixer"

_electronics, doi:10.3390/electronics9010161_

Round 1

Reviewer 1 Report

Generally speaking authors have demonstrated an interesting and important work on THz circuit and system, with both original idea well formulated and proved through fabrication and experiments. The virtue is high and therefore publication is recommended. 

Author Response

Thanks to the Reviewer's comments.  

Reviewer 2 Report

Dear Authors,

In my opinion, the manuscript presents novel and interesting results and would be of interest for the THz community and readers of Electronics.

However, due to some drawbacks, my overall recommendation is “major revision”:

I would suggest to change the title to better correspond to the presented work “Design, simulations and measurements of a 0.67 THz biased sub-harmonic mixer” or “Design and measurements of a 0.67 THz biased sub-harmonic mixer” Can you please underline in Introduction a novelty of your design. It is not clear where the mixer was fabricated. Is this your own work or you bought it? Can you please compare the simulation results with the experiments in term of conversion loss. It seems that there is a few dB gap. What about the isolation? It was simulated but not measured. Why? You put Table 3 without any comments. Can you please comment what is your mixer designed in comparison to other works. There are no corrects references – “Error! Reference source not found”     Can you please extend the discussion section underlining what are the advantages and disadvantages of your design; is there any room for improvements; what can be done better? Technical issues: Please check the manuscript thoroughly and correct all spelling mistakes, like silulator (113), boding wire (Fig. 2), in terahertz (100). Please improve quality of Fig.4. Line 76 – use subscripts and italics in the equation,  

Reviewer 3 Report

The paper presents a significant advancement with respect to the state-of-the art in the specific field. Therefore, the paper has merit and deserves consideration for publication in this Journal.

However, in order to be acceptable for publication, the paper should undergo a careful revision based on the considerations reported below:

The Introduction section should include, at the end, an information about the structure of the other sections of the paper The Discussion is not well structured, since it has no indication about the comparison with other existing studies on the relevant topic The Conclusion section should be more informative, since in this version it has no mention of possible practical applications of the solution described in the paper Overall, several typos and grammar/language errors are present and should be carefully corrected.

Round 2

Reviewer 2 Report

Dear Authors,

I think that currently the  manuscript meet the requirements and can be published.

Thank you for cooperation

Reviewer 3 Report

All my concerns were successfully answered.